# Chronic atrophic gastritis and intestinal metaplasia surrounding diffuse-type gastric cancer: Are they just bystanders in the process of carcinogenesis?

**Seung Yong Shin[1,2], Jie-Hyun Kim[2]\*, Jaeyoung Chun[2], Young Hoon Yoon[2], Hyojin Park[2]**

**1** Department of Internal Medicine, Chung-Ang University College of Medicine, Seoul, Korea, **2** Department of Internal Medicine, Gangnam Severance Hospital, Yonsei University College of Medicine, Seoul, Korea

\* otilia94@yuhs.ac

## Abstract

### Background

Gastric cancer (GC) is categorized as diffuse- and intestinal-type adenocarcinoma. Intestinal-type GC is associated with chronic gastritis, atrophic gastritis (AG), and intestinal metaplasia (IM), precursors of dysplastic changes. Diffuse-type GC is generally known to undergo *de novo* carcinogenesis and is not associated with chronic mucosal changes. However, clinically, AG and IM are frequently observed surrounding diffuse-type GC. This study aimed to evaluate the role of AG and IM in diffuse-type GC.

### Methods

We retrospectively reviewed the data of patients undergoing surgery for early GC. We divided patients with diffuse-type GC into two groups according to the presence of AG and IM based on Kyoto classification of gastritis. The clinicopathological characteristics were compared between the groups.

### Results

Among patients with diffuse-type GC, 52.5% patients had AG and 18.4% had severe AG. With regard to IM, 42.1% patients had IM and 17.1% had severe IM. Diffuse-type GC combined with severe AG or IM showed larger tumor size and higher submucosal invasion rate than that without severe AG or IM. However, the lymph node metastasis (LNM) rate was not significantly different between the two groups. In multivariate analysis, severe AG or IM was not an independent risk factor for LNM.

### Conclusions

Severe AG or IM surrounding diffuse-type gastric cancer suggests a collapse of normal mucosal barriers and leads to the spread of cancer cells. Although the association between

**Data Availability Statement:** All relevant data are within the manuscript and its Supporting Information files.

**Funding:** This study was financially supported by the "SEBANG" Faculty Research Assistance Program of Yonsei University College of Medicine (6-2014-0191). This research was supported by Basic Science Research Program through the National Research Foundation of Korea (NRF) funded by the Ministry of Education, Science and Technology (2018R1A2B6008139).

**Competing interests:** The authors have declared that no competing interests exist.

chronic mucosal changes and LNM is unclear, more caution is needed during endoscopy especially for complete resection of diffuse-type GC with these features.

## Introduction

Gastric cancer (GC) is classified into two major histological subtypes, diffuse-type and intestinal-type adenocarcinoma, according to the Lauren classification [1]. The differences in epidemiology, pathogenesis, and morphological appearance between these two GC types have been widely accepted [1,2]. Diffuse-type GCs are often reported in young patients, are known to be highly metastatic, and show more rapid progression and poor prognosis when compared with intestinal-type GCs [3,4]. In terms of carcinogenesis, intestinal-type GC is more likely to be associated with *Helicobacter pylori* (*H. pylori*) infection. The stepwise mucosal changes initiated by *H. pylori* infection are considered to be a major step in the carcinogenic cascade of intestinal-type GC [5,6], and atrophic gastritis (AG) and intestinal metaplasia (IM) are suggested to be the typical chronic mucosal changes in this cascade [7–9]. AG is characterized by multifocal loss of the original gastric glands, including mucus-secreting glands comprising parietal and chief cells. AG has been suggested to be the first histopathological lesion in intestinal-type GC [10,11]. IM is characterized by the appearance of glands with an intestinal phenotype that can be combined with multifocal atrophies involving the gastric mucosa. As atrophic and metaplastic glands replace the normal gastric glands, the number of normal gastric glands decreases, and precancerous processes such as dysplasia are initiated.

Carcinogenesis in diffuse-type GC is different from that in intestinal-type GC. A defined series of preneoplastic mucosal changes or precancerous lesions are not usually found in diffuse-type GC. Less well differentiated, non-glandular formations, with an occasional presence of signet ring cells, are the pathological characteristics of diffuse-type GC [12]. However, it is relatively common to find background mucosa in association with AG or IM surrounding diffuse-type GC in a clinical setting during diagnostic endoscopy (Fig 1). Some diffuse-type GCs are surrounded by these chronic mucosal changes, similar to that in intestinal-type GC. Although pathological evaluation was considered to be the gold standard diagnostic modality for AG and IM, recent studies have reported that endoscopic diagnosis of these chronic mucosal changes is also useful for identifying patients at risk of developing GC [13,14]. As chronic mucosal changes account for a large proportion of cases of carcinogenesis in intestinal-type GC, we wished to investigate the implications of chronic mucosal changes surrounding diffuse-type GC noted during endoscopy. It is necessary to determine how these chronic mucosal changes, which are precancerous lesions in intestinal-type GC, affect progression in diffuse-type GC for further understanding of carcinogenesis in diffuse-type GC. Therefore, this study aimed to identify the role of AG and IM surrounding diffuse-type GC.

## Methods

### Study design

This study evaluated 808 consecutive patients who underwent surgery for the treatment of early gastric cancer (EGC) at Gangnam Severance Hospital, Seoul, Korea. We collected the medical records and endoscopic images of patients who were diagnosed with EGC. Patients without preoperative endoscopic images were excluded from this study. Furthermore, patients in whom it was difficult to evaluate AG or IM owing to low resolution or insufficient images were excluded. Finally, 451 patients with diffuse-type GC were enrolled. The intestinal-type

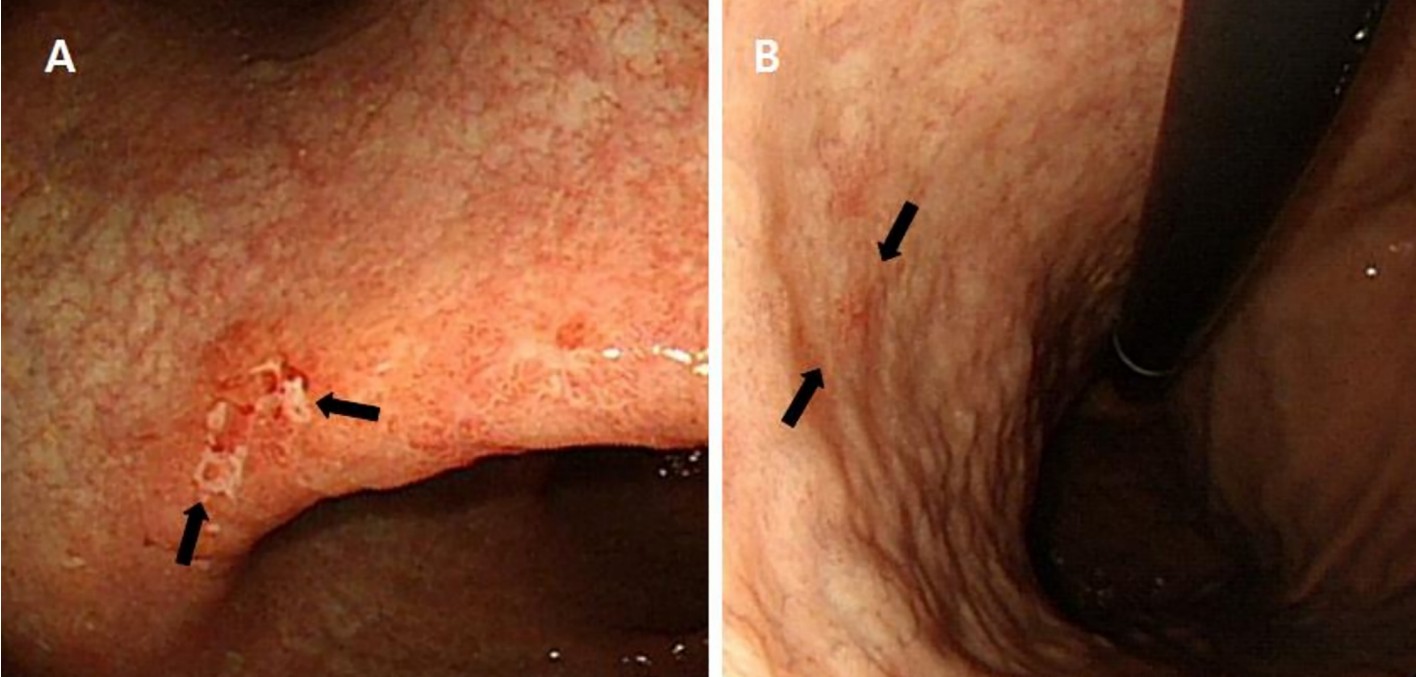

**Fig 1. Clinical cases of pathologically confirmed diffuse-type early gastric cancer (arrow).** On endoscopic evaluation, (A) atrophic gastritis and (B) intestinal metaplasia are observed in the surrounding mucosa.

GC group was randomly selected from the remaining study cohort to compare their clinico-pathological features with those of the diffuse-type GC group. We analyzed the clinicopatho-logical data and endoscopic images. This study protocol was approved by the Institutional Review Board of Gangnam Severance Hospital (3-2019-0006) and was carried out in accor-dance with the 1964 Declaration of Helsinki and its later amendments. Informed consent was not required as this study was a retrospective analysis of existing administrative and clinical data.

## Endoscopic image review

All endoscopic images with unanalyzed clinicopathological data of patients enrolled during the study period were collected and reanalyzed. AG and IM were categorized according to the Kyoto classification of gastritis, proposed at the 85th Congress of the Japan Gastroenterological Endoscopy Society for standardizing endoscopic findings of gastritis [15]. This classification allows grading of endoscopically visible AG and IM. AG is assessed by the Kimura-Takemoto classification (Close (C)-1, C-II, C-II; Open (O)-I, O-II, and O-III), and classified into three grades (none, C0-C1; mild, CII-CIII; and severe, OI-OIII)[16]. IM is also classified into three grades (none: none, mild: within the antrum, and severe: up to the corpus). In our study, two experienced gastrointestinal endoscopists re-examined all endoscopic images in the same manner and categorized AG and IM. They arrived at a consensus on this categorization through regular meetings.

## Clinicopathological evaluation

The medical records of patients, including their clinicopathological characteristics, were reviewed by specialists who were blinded to the endoscopic findings. Pathological review was

performed based on reports of surgical specimens. The gross appearance of the tumors was classified using the criteria of the Japanese Gastric Cancer Association, and the tumors were classified into two groups accordingly. Protruding type (Type I) and superficially elevated type (Type IIa) tumors were classified as elevated and the other types of tumor as non-elevated. Longitudinal tumor location was defined as being in the upper (fundus, cardia, and upper body), middle (mid-body, lower body, and the angle), and lower (antrum, prepylorus, and pylorus) stomach. Tumor size was measured using the longest and shortest diameters. The depth of tumor invasion was categorized as mucosal invasion (M), submucosal invasion of 500 μm or less from the muscularis mucosae (SM1), and submucosal invasion of more than 500 μm from the muscularis mucosae (SM2). Lymphovascular invasion (LVI), perineural invasion (PNI), and lymph node metastasis (LNM) were also identified. The clinicopathological features were compared between the severe AG or IM group and none-to-moderate AG or IM group.

## Statistical analysis

For categorical variables, a Pearson chi-square test was used. Student t-test was used to compare the means of continuous variables, and continuous variables are presented as the mean ± standard deviation (SD). Variables that were significant in univariate analysis were subsequently tested by multivariate logistic regression analysis. Irrespective of the results of the univariate analysis, variables that could potentially influence the results were included in multivariate analysis. For all comparisons, two-sided $P$-values $<0.05$ were considered statistically significant. Statistical analyses were performed using SPSS version 18.0 (SPSS, Chicago, IL).

## Results

### Clinicopathological characteristics of intestinal-type and diffuse-type gastric cancer

Table 1 summarizes the clinicopathological characteristics of intestinal- and diffuse-type GCs. Diffuse-type GC, defined according to Lauren classification, included poorly differentiated adenocarcinoma (n = 204, 45.3%), signet ring cell carcinoma (SRC) (n = 245, 54.3%), and mucinous adenocarcinoma (n = 2, 0.4%). Patients with diffuse-type GC were younger and included a higher proportion of females than patients with intestinal-type GC. Intestinal-type GC showed a more elevated appearance and was more frequently located in the lower stomach. Severe IM was identified significantly more often in intestinal-type GC patients (30.5% vs. 17.3%, $P < 0.001$). Among patients with diffuse-type GC, 52.5% had AG and 18.4% had severe AG. In terms of IM, 42.1% patients had IM and 17.1% had severe IM in the group with diffuse-type GC. Diffuse-type GC was more commonly found to be confined to the mucosal layer in contrast to intestinal-type GC. LVI, PNI, and LNM rates were not significantly different between the two types of GCs.

### Comparison between diffuse-type gastric cancer with and without severe atrophic gastritis

According to the severity of AG defined by Kyoto classification, patients with diffuse-type GC were divided into the severe and non-severe AG groups. Table 2 shows the clinicopathological features of the two groups. Diffuse-type GC with severe AG was frequently found in older patients. The tumor size was larger (longest diameter, 3.671 ± 2.543 cm vs. 2.659 ± 1.762, $P = 0.001$; shortest diameter, 2.551 ± 1.654 cm vs. 1.948 ± 1.740 cm, $P = 0.006$) and the rates of SM invasion and LVI were higher in the severe AG group than in the non-severe AG group.

**Table 1. Clinicopathological characteristics of intestinal- and diffuse-type gastric cancers.**

| | Diffuse-type GC (n = 451) | Intestinal-type GC (n = 210) | P value |
|---|---|---|---|
| Age (years, mean ± SD) | 56.88 ± 12.49 | 62.94 ± 10.83 | <0.001 |
| Male (n, %) | 229 (50.8) | 151 (71.9) | <0.001 |
| Pathological findings (n, %) | | | |
| Well-differentiated adenocarcinoma | | 99 (47.1) | |
| Moderately differentiated adenocarcinoma | | 111 (52.9) | |
| Poorly differentiated adenocarcinoma | 204 (45.3) | - | |
| Signet ring cell carcinoma | 245 (54.3) | - | |
| Mucinous adenocarcinoma | 2 (0.4) | - | |
| Kyoto classification | | | 0.659 |
| Atrophic gastritis (n, %) | | | 0.528 |
| None to mild | 368 (81.6) | 167 (79.5) | |
| Severe | 83 (18.4) | 43 (20.5) | |
| Intestinal metaplasia (n, %) | | | <0.001 |
| None to mild | 373 (82.7) | 146 (69.5) | |
| Severe | 78 (17.3) | 64 (30.5) | |
| *H. pylori* infection (n, %) | 206 (45.8) | 152 (73.4) | <0.001 |
| Longitudinal location (n, %) | | | 0.009 |
| Lower | 272 (60.3) | 150 (71.4) | |
| Mid | 141 (31.3) | 42 (20.0) | |
| Upper | 38 (8.4) | 18 (8.6) | |
| Gross appearance (n, %) | | | <0.001 |
| Elevated | 67 (14.9) | 62 (29.5) | |
| Non-elevated | 384 (85.1) | 148 (70.5) | |
| Depth of invasion (n, %) | | | 0.031 |
| M | 259 (57.4) | 99 (47.1) | |
| SM1 (≤500 μm) | 40 (8.9) | 28 (13.3) | |
| SM2 (≥500 μm) | 152 (33.7) | 83 (39.5) | |
| LVI (n, %) | 55 (12.2) | 16 (7.6) | 0.075 |
| PNI (n, %) | 15 (3.3) | 5 (2.4) | 0.509 |
| LNM (n, %) | 38 (8.4) | 9 (4.3) | 0.054 |

*H. pylori*, *Helicobacter pylori*; M, mucosa; SM, submucosa; LVI, lymphovascular invasion; PNI, perineural invasion; LNM, lymph node metastasis; GC, gastric cancer

However, the LNM rate was not significantly different between the groups. In multivariate analysis, severe AG was not an independent risk factor for LNM (Table 3).

## Comparison between diffuse-type gastric cancer with and without severe intestinal metaplasia

Patients with diffuse-type GC were also divided according to the severity of IM as defined by Kyoto classification into the severe and non-severe IM groups (Table 4). Upon comparing the two groups, older age and male sex were commonly associated with severe IM. The tumor size was larger in the severe IM group than in the non-severe IM group (longest diameter, 3.488 ± 2.657 cm vs. 2.725 ± 1.764 cm, P = 0.002; shortest diameter (cm), 2.449 ± 1.628 cm vs 1.973 ± 1.752 cm, P = 0.032). However, the depth of invasion, LVI, PNI, and LNM showed no significant differences between the groups. In multivariate analysis, severe IM was not found to be an independent risk factor for LNM (Table 3).

**Table 2. Comparison between diffuse-type gastric cancer with and without severe atrophic gastritis.**

| | Without severe atrophic gastritis (n = 368) | With severe atrophic gastritis (n = 83) | P value |
|---|---|---|---|
| Age (years, mean ± SD) | 53.17 ± 12.17 | 63.29 ± 10.51 | <0.001 |
| Male (n, %) | 183 (49.7) | 46 (55.4) | 0.349 |
| *H. pylori* infection (n, %) | 125 (33.9) | 27 (32.5) | 0.307 |
| Longitudinal location (n, %) | | | 0.354 |
| Lower | 219 (59.5) | 53 (63.9) | |
| Mid | 120 (32.6) | 21 (25.3) | |
| Upper | 29 (7.9) | 9 (10.8) | |
| Gross appearance (n, %) | | | 0.362 |
| Elevated | 52 (14.1) | 15 (18.1) | |
| Non-elevated | 316 (85.9) | 68 (81.9) | |
| Depth of invasion (n, %) | | | 0.001 |
| M | 225 (61.1) | 34 (41.0) | |
| SM | 143 (38.9) | 49 (59.0) | |
| Size (cm, mean ± SD) | | | |
| Longest diameter | 2.659 ± 1.762 | 3.671 ± 2.543 | 0.001 |
| Shortest diameter | 1.948 ± 1.740 | 2.551 ± 1.654 | 0.006 |
| LVI (n, %) | 34 (9.3) | 21 (25.3) | <0.001 |
| PNI (n, %) | 10 (2.7) | 5 (6.0) | 0.129 |
| LNM (n, %) | 30 (8.2) | 8 (9.6) | 0.660 |

*H. pylori*, *Helicobacter pylori*; M, mucosa; SM, submucosa; LVI, lymphovascular invasion; PNI, perineural invasion; LNM, lymph node metastasis

**Table 3. Multivariate analysis of clinicopathological characteristics associated with LNM in diffuse-type gastric cancer.**

| | Odds ratio (95% confidence interval) | P value |
|---|---|---|
| Severe atrophic gastritis | | 0.069 |
| No | Reference | |
| Yes | 0.381 (0.134–1.080) | |
| Severe intestinal metaplasia | | 0.833 |
| No | Reference | |
| Yes | 1.112 (0.416–2.975) | |
| Tumor size | | 0.023 |
| ≤2 cm | Reference | |
| >2 cm | 2.985 (1.165–7.647) | |
| Depth of invasion | | 0.004 |
| M | Reference | |
| SM | 5.586 (1.576–17.803) | |
| LVI | | <0.001 |
| No | Reference | |
| Yes | 10.073 (4.219–24.053) | |
| PNI | | 0.229 |
| No | Reference | |
| Yes | 2.237 (0.602–8.318) | |

LNM, lymph node metastasis; M, mucosa; SM, submucosa; LVI, lymphovascular invasion; PNI, perineural invasion

**Table 4. Comparison between diffuse-type gastric cancer with and without severe intestinal metaplasia.**

| | Without severe intestinal metaplasia (n = 373) | With severe intestinal metaplasia (n = 78) | P value |
|---|---|---|---|
| Age (years, mean ± SD) | 53.83 ± 12.43 | 60.73 ± 11.32 | <0.001 |
| Male (n, %) | 178 (47.8) | 49 (63.6) | 0.012 |
| *H. pylori* infection (n, %) | 127(34.0) | 25 (32.1) | 0.191 |
| Longitudinal location (n, %) | | | 0.648 |
| Lower | 226 (60.5) | 46 (58.4) | |
| Mid | 114 (3.6) | 27 (35.1) | |
| Upper | 33 (8.9) | 5 (6.5) | |
| Gross appearance (n, %) | | | 0.858 |
| Elevated | 55 (14.8) | 12 (15.6) | |
| Non-elevated | 318 (85.2) | 66 (84.4) | |
| Depth of invasion (n, %) | | | 0.263 |
| M | 219 (58.9) | 40 (51.9) | |
| SM | 154 (41.1) | 38 (59.0) | |
| Size (cm, mean ± SD) | | | |
| Longest diameter | 2.725 ± 1.764 | 3.488 ± 2.657 | 0.002 |
| Shortest diameter | 1.973 ± 1.752 | 2.449 ± 1.628 | 0.032 |
| LVI (n, %) | 40 (10.8) | 15 (19.5) | 0.054 |
| PNI (n, %) | 12 (3.2) | 3 (3.9) | 0.766 |
| LNM (n, %) | 29 (7.8) | 9 (11.7) | 0.264 |

*H. pylori*, *Helicobacter pylori*; M, mucosa; SM, submucosa; LVI, lymphovascular invasion; PNI, perineural invasion; LNM, lymph node metastasis

## Discussion

In accordance with previous studies, diffuse-type GC was associated with younger age and female sex compared with intestinal-type GC in our study [17,18]. Older patients and IM were significantly more common among those with intestinal-type GC. AG was also more frequently observed in intestinal-type GC, although the difference was not statistically significant. Intestinal-type GC showed more elevated lesions and was commonly located at a lower location, which is in accordance with previous findings [17]. Our study was unable to demonstrate a higher rate of SM invasion in diffuse-type GC than in intestinal-type GC. A possible explanation for this might be the relatively higher proportion (54.3%) of SRC cases in our study, which is known to be associated with a low incidence of SM invasion [18].

Severe AG and IM were noted in approximately 20% patients with diffuse-type GC during diagnostic endoscopy. We limited the comparison to that between severe AG or IM and others to minimize interobserver or intraobserver variation during endoscopic evaluation of AG or IM. Older patients more commonly showed these chronic mucosal changes. Chronic mucosal changes resulting from *H. pylori* infection are involved in the carcinogenesis of intestinal-type GC. In this study, patients with intestinal-type GC showed a higher infection rate of *H. pylori* compared to those with diffuse-type GC. *H. pylori* infection induces a chronic inflammatory reaction in the gastric mucosa, which causes atrophic changes or transformation of the glandular gastric epidermis to intestinal-type epithelium. The alkaline environment caused by these chronic mucosal changes facilitates the proliferation of *H. pylori*, which in turn leads to cell turnover increase and accumulation of mitotic errors resulting in cancerous changes of the gastric mucosa [19–21]. The higher infection rate of *H. pylori* in intestinal-type GC is thought

to be associated with this carcinogenic cascade. Diffuse-type GC has widely been believed to arise from the mucosa without chronic mucosal changes, and shows no precancerous lesions. Therefore, endoscopists tend to presumably diagnose cancerous lesions with background chronic mucosal changes as intestinal-type GC based on diagnostic endoscopy. However, our study showed that cases with diffuse-type GC arising from background mucosa with AG or IM are more common than expected. A previous Japanese study demonstrated that endoscopic findings in patients with undifferentiated adenocarcinoma diagnosed according to the Japanese Classification of Gastric Carcinoma (14[th] edition) showed the presence of severe AG in 21.8% and severe IM in 19.4% patients [22]. They also suggested ambiguous glandular duct architecture, venous invasion, and reddish lesions as characteristic endoscopic findings in undifferentiated adenocarcinomas with severe AG [22]. Another study evaluating the background mucosa in patients with GC during diagnostic endoscopy found that open-type AG according to the Kimura-Takemoto classification is present in many patients with undifferentiated adenocarcinoma [23]. Our findings support the results of these previous studies. The incidence of severe AG and IM in diffuse type-GC may be explained by the increasing prevalence of AG and IM with age, as well as *H. pylori* infection [24], as the mean age of patients with severe AG or IM was higher than that of patients with non-severe AG or IM in our study. Thus, not only intestinal-type GC but also diffuse-type GC can arise from background mucosa with severe AG or IM. Therefore, a more careful and non-prejudiced approach is needed for evaluating lesions with these chronic mucosal changes during endoscopy. However, there is no previous study on whether these chronic mucosal changes are associated with aging or play any meaningful role in diffuse-type GC. Thus, we attempted to investigate the role of chronic mucosal changes surrounding diffuse-type GC in this study.

Compared with the non-severe AG group, the severe AG group showed a larger tumor size and higher SM invasion and LVI rates. The tumor size was also larger in the severe IM group compared to that in the non-severe IM group. A previous study demonstrated that the tumor size is larger in patients with undifferentiated-type GC with severe AG than in those with undifferentiated-type GC with mild-to-moderate AG [22]. The reason for the increase in tumor size when associated with severe AG or IM is uncertain. However, cancer cells seem to spread more easily in both horizontal and vertical directions when the background mucosa shows AG or IM. A possible explanation for these pathological findings is assumed to be the fact that the surrounding mucosa acts as a mechanical barrier to the spread of cancer cells in diffuse-type GC.

In terms of the barrier function of the mucosa, previous studies have reported that a diverse group of transmembrane proteins maintain the epithelial barrier function against pathogens. Tight junction proteins are present on the apical end of the lateral membrane surface in columnar epithelial cells and function as a barrier [25,26]. Although scarce data exist on the physiology or function of this barrier, barrier dysfunction is suggested to be a risk factor for cancer development in association with *H. pylori* infection [27]. The concept of mechanical barriers related to the spreading of cancer cells was reported in a study on SRC [28]. The surrounding mucosal pattern differs according to the manner in which the tumor spreads, and cells in SRC tend to spread diffusely into the subepithelial area in the presence of background mucosa with AG or IM. The authors further suggested that if the mechanical barrier represented by the surrounding mucosa is weak, such as in cases with AG or IM, cancer cells tend to spread sporadically or diffusely into the deeper layers of the mucosa. In addition, the status of the surrounding mucosa is suggested to be a predictive factor of intramucosal spreading patterns in SRC. In line with this, our study demonstrated that the disrupted mucosal barriers, as seen in cases with severe AG or IM, allow the cancer cells in diffuse-type GC to spread more easily in both horizontal and vertical directions. The fact that metachronous cancer occurs

more frequently in cases with diffuse-type GC is also thought to be associated with AG or IM [29]. AG or IM, which suggests weakening of the mechanical barrier, can facilitate sporadic spreading of cancer cells. Thus, more careful assessment is required in patients with diffuse-type GC with severe AG or IM, especially when performing endoscopic resection as well as diagnostic endoscopy.

One interesting finding of our study is that the LNM rate showed no significant differences between the severe AG or IM and non-severe groups. A previous study investigating SRC showed similar results to those shown in our study [28]. Although the weakening of the mechanical barrier induced by AG or IM combined with diffuse-type GC is associated with the spreading of cancer cells into the mucosa and allows the tumor to increase in size or show LVI, it does not seem to have a significant effect on the prognosis of patients. Severe AG or IM affects the tumor size and SM invasion; however, it was not an independent risk factor of LNM. Although the exact reason for this finding remains unclear, a possible explanation for this might be that AG and IM themselves do not directly affect LNM, but the risk of LNM is increased owing to changes in the spreading pattern of cancer cells caused by these chronic mucosal changes. In other words, AG or IM in diffuse-type GC do not influence the aggressive behavior of cancer, but rather provide an environment in which the cancer cells spread more aggressively. Validation of our findings with a large-scale study is needed to gain further insight.

A previous Japanese study demonstrated that most intestinal-type GCs occur in association with severe AG or IM, whereas diffuse-type GCs occur predominantly at the atrophic border which is characterized by active inflammation. These authors suggested that active inflammation may cause DNA damage in gastric epithelial cells [30]. Several studies have also suggested genetic abnormalities in cell adhesion factors, rather than chronic mucosal changes, as the cause of diffuse-type GC [31–34]; our study supports this notion. In contrast to its role as a precancerous lesion in the carcinogenesis of intestinal-type GCs, AG or IM in diffuse-type GCs may play a role related to the mechanical barrier function of the mucosa, which is not associated with carcinogenesis.

This study has several limitations. Firstly, it was a retrospective, single-center study. Secondly, interobserver bias may exist given that this study was performed based on a review of endoscopic images. However, endoscopists applied the same criteria for the endoscopic review and reached consensus through periodic meetings. To the best of our knowledge, this is the first study to evaluate the role of AG or IM in diffuse-type GC. Further large-scale studies are needed to validate our results.

In conclusion, AG or IM surrounding diffuse-type GC suggests the destruction of the normal mechanical barrier that prevents the spread of cancer cells. Although these chronic mucosal changes may not play a role in the process of carcinogenesis in diffuse-type GC, in contrast to their role in intestinal-type GC, more careful assessment is needed in cases with diffuse-type GC during endoscopy, especially in order to achieve a complete endoscopic resection.

## Supporting information

**S1 File. Data set.**
(XLSX)

## Acknowledgments

The English in this document has been checked by at least two professional editors, both native speakers of English (www.editage.co.kr).

## Author Contributions

**Conceptualization:** Jie-Hyun Kim.

**Data curation:** Seung Yong Shin.

**Formal analysis:** Seung Yong Shin.

**Investigation:** Seung Yong Shin.

**Methodology:** Seung Yong Shin.

**Project administration:** Seung Yong Shin.

**Resources:** Seung Yong Shin, Jaeyoung Chun, Young Hoon Yoon, Hyojin Park.

**Software:** Seung Yong Shin.

**Supervision:** Jie-Hyun Kim.

**Validation:** Seung Yong Shin.

**Writing – original draft:** Seung Yong Shin.

**Writing – review & editing:** Jie-Hyun Kim, Jaeyoung Chun, Young Hoon Yoon, Hyojin Park.

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
