## [Decision Letter · Decision Letter 0]

30 Sep 2019

PONE-D-19-16075

Chronic atrophic gastritis and intestinal metaplasia surrounding diffuse-type gastric cancer: Are they just bystanders in the process of carcinogenesis?

PLOS ONE

Dear Dr. Kim,

Thank you for submitting your manuscript to PLOS ONE. After careful consideration, we feel that it has merit but does not fully meet PLOS ONE’s publication criteria as it currently stands. Therefore, we invite you to submit a revised version of the manuscript that addresses the points raised during the review process.

We would appreciate receiving your revised manuscript by Nov 14 2019 11:59PM. To enhance the reproducibility of your results, we recommend that if applicable you deposit your laboratory protocols in protocols.io, where a protocol can be assigned its own identifier (DOI) such that it can be cited independently in the future. For instructions see: http://journals.plos.org/plosone/s/submission-guidelines#loc-laboratory-protocols

We look forward to receiving your revised manuscript.

Kind regards,

Dajun Deng, M.D.

Academic Editor

PLOS ONE

Journal Requirements:

2. In your ethics statement in the manuscript and in the online submission form, please provide additional information about the patient records used in your retrospective study. Specifically, please ensure that you have discussed whether all data were fully anonymized before you accessed them and/or whether the IRB or ethics committee waived the requirement for informed consent. If patients provided informed written consent to have data from their medical records used in research, please include this information.

Additional Editor Comments (if provided):

Reviewers' comments:

Reviewer's Responses to Questions

**Comments to the Author**

1. Is the manuscript technically sound, and do the data support the conclusions?

Reviewer #1: Yes

2. Has the statistical analysis been performed appropriately and rigorously? 

Reviewer #1: Yes

3. Have the authors made all data underlying the findings in their manuscript fully available?

Reviewer #1: Yes

4. Is the manuscript presented in an intelligible fashion and written in standard English?

Reviewer #1: Yes

5. Review Comments to the Author

Reviewer #1: The manuscript by Shin et al entitled “Chronic atrophic gastritis and intestinal metaplasia surrounding diffuse-type gastric cancer: Are they just bystanders in the process of carcinogenesis?” evaluated the role and meaning of atrophic gastritis (AG) and intestinal metaplasia (IM) in diffuse-type GC. They demonstrated that diffuse-type GC combined with severe AG or IM showed larger tumor size and higher submucosal invasion rate than that without severe AG or IM. But the multivariate analysis showed that severe AG or IM was not an independent risk factor for LNM. Overall, this topic is interesting. I have the following minor concerns:

1. In Table 1, the authors mentioned that the intestinal type gastric cancer had higher infection rates with H. pylori. Can the authors include some studies in the discussion to explore the possible mechanism?

2. Diffuse-type GC combined with severe AG or IM showed larger tumor size and higher submucosal invasion rate than that without severe AG or IM. But the multivariate analysis showed that severe AG or IM was not an independent risk factor for LNM. What do the authors think might be the reasons? In Line 244, the authors stated that “Although the weakening of mechanical barrier induced by AG or IM combined with diffuse-type GC is associated with the spreading of cancer cells into the mucosa and allows the tumor to increase in size or to invade LVI, it does not seem to have a significant effect on the prognosis of patients.” But they did not explain this phenomenon.

3. In Line 247, the authors stated that “In contrast to its role as a precancerous lesion in the carcinogenesis of intestinal-type GCs, AG or IM in diffuse-type GCs may play a role related to a mechanical barrier, which is not associated with carcinogenesis.” Do the authors mean that AG or IM in diffuse-type GCs can destroy the normal barrier to protect the spread of tumor cells? Meanwhile, can the authors provide some evidences to demonstrated that AG or IM in diffuse-type GCs is not associated with carcinogenesis?

4. In Line 257, the authors stated that “In conclusion, AG or IM surrounding diffuse-type GC suggests the destruction of normal mechanical barriers that protect the spread of cancer cells.” But in Table 2, the result showed that with or without severe AG was not associated with LNM rates. Why?

5. English editing is recommended.

6. PLOS authors have the option to publish the peer review history of their article (what does this mean?). If published, this will include your full peer review and any attached files.

Reviewer #1: No

---

## [Author Response · Author response to Decision Letter 0]

16 Nov 2019

Reviewer #1 

The manuscript by Shin et al entitled “Chronic atrophic gastritis and intestinal metaplasia surrounding diffuse-type gastric cancer: Are they just bystanders in the process of carcinogenesis?” evaluated the role and meaning of atrophic gastritis (AG) and intestinal metaplasia (IM) in diffuse-type GC. They demonstrated that diffuse-type GC combined with severe AG or IM showed larger tumor size and higher submucosal invasion rate than that without severe AG or IM. But the multivariate analysis showed that severe AG or IM was not an independent risk factor for LNM. Overall, this topic is interesting. I have the following minor concerns.

1. In Table 1, the authors mentioned that the intestinal type gastric cancer had higher infection rates with H. pylori. Can the authors include some studies in the discussion to explore the possible mechanism?

Response> Thank you for your valuable comments. As reviewer commented, we have included some studies in the discussion to explore the possible mechanisms. The higher infection rate of H. pylori in intestinal-type GC is thought to be associated with carcinogenesis of GC. Previous studies have suggested that the chronic inflammatory reactions to the gastric mucosa caused by H. pylori infection induce atrophic changes or mucosal transformation to intestinal-type epithelium, and facilitate the proliferation of H. pylori in an environment of hypochlorhydria. When this process continues, cell turnover increases and mitotic errors accumulates which result in cancerous changes of gastric mucosa. We presented the additional comments in the discussion. The details of the changes are as follows: 

(Discussion section, page 15, line 214): “In this study, patients with intestinal-type GC showed a higher infection rate of H. pylori compared to those with diffuse-type GC. H. pylori infection induces a chronic inflammatory reaction in the gastric mucosa, which causes atrophic changes or transformation of the glandular gastric epidermis to intestinal-type epithelium. The alkaline environment caused by these chronic mucosal changes facilitates the proliferation of H. pylori, which in turn leads to cell turnover increase and accumulation of mitotic errors resulting in cancerous changes of the gastric mucosa [19-21]. The higher infection rate of H. pylori in intestinal-type GC is thought to be associated with this carcinogenic cascade.”

2. Diffuse-type GC combined with severe AG or IM showed larger tumor size and higher submucosal invasion rate than that without severe AG or IM. But the multivariate analysis showed that severe AG or IM was not an independent risk factor for LNM. What do the authors think might be the reasons? In Line 244, the authors stated that “Although the weakening of mechanical barrier induced by AG or IM combined with diffuse-type GC is associated with the spreading of cancer cells into the mucosa and allows the tumor to increase in size or to invade LVI, it does not seem to have a significant effect on the prognosis of patients.” But they did not explain this phenomenon.

Response> Thank you for your valuable comments. The aim of this study was to determine whether the presence of AG or IM in the diffuse-type GC affects the biologic behavior such as LNM. The results showed that the diffuse-type GC surrounded by AG or IM had larger tumor size and deeper invasion depth, but in multivariate analysis, AG or IM did not affect LNM. Therefore, it can be interpreted that the increased risk of LNM is not attributed to AG or IM itself, but to the changed spreading pattern caused by AG or IM. AG or IM destroys the normal mechanical barrier and allows cancer cells to spread more easily in both horizontal and vertical direction which leads to increasing LNM risk. In other words, the meaning of AG or IM in diffuse-type GC is not itself affecting the aggressive behavior of cancer, but rather providing an environment in which the cancer cells more aggressively spread. However, validation with large scale study is needed. We presented the additional comments regarding this concern in the discussion. The details of the changes are included in the answers to the question 4. 

3. In Line 247, the authors stated that “In contrast to its role as a precancerous lesion in the carcinogenesis of intestinal-type GCs, AG or IM in diffuse-type GCs may play a role related to a mechanical barrier, which is not associated with carcinogenesis.” Do the authors mean that AG or IM in diffuse-type GCs can destroy the normal barrier to protect the spread of tumor cells? Meanwhile, can the authors provide some evidences to demonstrated that AG or IM in diffuse-type GCs is not associated with carcinogenesis?

Response > Thank you for your valuable comments. As we mentioned in discussion section, barrier dysfunction of normal mucosa is suggested to be a risk factor for cancer development in association with H. pylori infection, and the concept of mechanical barriers related to the spreading of cancer cells was reported in a previous study of SRC. We also have suggested the role of normal mucosa as a mechanical barrier for protecting tumor cell spread. Chronic mucosal changes are the destruction of normal glandular structures which means the collapse of mucosal barriers. Regarding evidences to demonstrate that AG or IM in diffuse-type GCs is not associated with carcinogenesis, we have provided evidences related to the differences of carcinogenesis between intestinal-type and diffuse-type GC. Although genetic factors, gastric environment, and Helicobacter pylori infection have been associated with the pathogenicity and development of intestinal-type GC that follows the Correa’s cascade, the pathogenicity of diffuse-type GC remains mostly unknown and undefined. Previous Japanese study showed that most intestinal-type GC occurs distal to the atrophic border, where severe atrophic gastric mucosa and intestinal metaplasia has been present for a longer period. However, they found that diffuse-type arises close to the atrophic border, and suggested that the active element of inflammation, which is predominantly in the region of the atrophic border, may damage DNA in gastric epithelial cells (Yoshimura T et al. Scand J Gastroenterol. 1999 Nov;34(11):1077-81). Although underlying molecular pathways have not yet been well-studied and understood, genetic abnormalities in the cell adherence factors such as E-cadherin, and cellular activities that cause impaired cell integrity and physiology have been suggested and documented as contributing factors of diffuse-type GC rather than chronic mucosal changes (Cho SY et al. Gastroenterology. 2017 Aug;153(2):536-549.e26, Ghatak S etl al. BMC Med Genet. 2017 Jun 2;18(1):61, Kakiuchi M et al. Nat Genet. 2014 Jun;46(6):583-7). We presented the additional comments regarding this concern in the discussion. The details of the changes are as follows: 

(Discussion section, page 18, line 290): “A previous Japanese study demonstrated that most intestinal-type GCs occur in association with severe AG or IM, whereas diffuse-type GCs occur predominantly at the atrophic border which is characterized by active inflammation. These authors suggested that active inflammation may cause DNA damage in gastric epithelial cells [30]. Several studies have also suggested genetic abnormalities in cell adhesion factors, rather than chronic mucosal changes, as the cause of diffuse-type GC [31-34]; our study supports this notion.”

4. In Line 257, the authors stated that “In conclusion, AG or IM surrounding diffuse-type GC suggests the destruction of normal mechanical barriers that protect the spread of cancer cells.” But in Table 2, the result showed that with or without severe AG was not associated with LNM rates. Why? 

Response > Thank you for your valuable comments. The possible explanation for this might be that AG and IM themselves do not directly affect the LNM, but the risk of LNM can be increased due to changes in the spreading patterns of cancer cell caused by these chronic mucosal changes. Cancer cells are able to spread more easily toward both the horizontal and vertical sides when the background mucosa shows AG or IM. Therefore, the role of AG and IM surrounding diffuse-type GC are limited to the destruction of the normal mechanical barrier, not affect directly on LNM or any process of carcinogenesis. However, careful assessments are required in AG and IM surrounding diffuse-type GC in that they affect the spreading patterns of the cancer cell which are directly associated with LNM. Further studies are warranted to better understand severe mucosal changes and LNM in diffuse-type GC. We presented the additional comments regarding this concern in the discussion. The details of the changes are as follows: 

(Discussion section, page 18, line 281): “Severe AG or IM affects the tumor size and SM invasion; however, it was not an independent risk factor of LNM. Although the exact reason for this finding remains unclear, a possible explanation for this might be that AG and IM themselves do not directly affect LNM, but the risk of LNM is increased owing to changes in the spreading pattern of cancer cells caused by these chronic mucosal changes. In other words, AG or IM in diffuse-type GC do not influence the aggressive behavior of cancer, but rather provide an environment in which the cancer cells spread more aggressively. Validation of our findings with a large-scale study is needed to gain further insight.”

5. English editing is recommended

Response> Thank you for your kind comment. The English has been checked by at least two professional editors. We presented the additional comments regarding English editing. 

(Page 20, Acknowledgements, line 341): “The English in this document has been checked by at least two professional editors, both native speakers of English.”

Thank you for all your valuable comments above. We think the editors’ comments have significantly improved the quality of our manuscript by guiding our insights appropriately.

---

## [Editor Report · Decision Letter 1]

27 Nov 2019

Chronic atrophic gastritis and intestinal metaplasia surrounding diffuse-type gastric cancer: Are they just bystanders in the process of carcinogenesis?

PONE-D-19-16075R1

Dear Dr. Kim,

We are pleased to inform you that your manuscript has been judged scientifically suitable for publication and will be formally accepted for publication once it complies with all outstanding technical requirements.

With kind regards,

Dajun Deng, M.D.

Academic Editor

PLOS ONE
---

## [Editor Report · Acceptance letter]

10 Dec 2019

PONE-D-19-16075R1 

Chronic atrophic gastritis and intestinal metaplasia surrounding diffuse-type gastric cancer: Are they just bystanders in the process of carcinogenesis? 

Dear Dr. Kim:

I am pleased to inform you that your manuscript has been deemed suitable for publication in PLOS ONE. Congratulations! Your manuscript is now with our production department. 

With kind regards,

on behalf of

Prof. Dajun Deng 

Academic Editor

PLOS ONE